# Identification of Tumor-Suppressive *miR-30e-3p* Targets: Involvement of *SERPINE1* in the Molecular Pathogenesis of Head and Neck Squamous Cell Carcinoma

**DOI:** 10.3390/ijms23073808

**Published:** 2022-03-30

**Authors:** Chikashi Minemura, Shunichi Asai, Ayaka Koma, Ikuko Kase-Kato, Nozomi Tanaka, Naoko Kikkawa, Atsushi Kasamatsu, Hidetaka Yokoe, Toyoyuki Hanazawa, Katsuhiro Uzawa, Naohiko Seki

**Affiliations:** 1Department of Oral Science, Graduate School of Medicine, Chiba University, Chiba 260-8670, Japan; minemura@chiba-u.jp (C.M.); axna4812@chiba-u.jp (A.K.); kato.ikuko@chiba-u.jp (I.K.-K.); n.tanaka@chiba-u.jp (N.T.); kasamatsua@faculty.chiba-u.jp (A.K.); uzawak@faculty.chiba-u.jp (K.U.); 2Department of Oral and Maxillofacial Surgery, National Defense Medical College Hospital, Tokorozawa 359-8513, Japan; yokoe@ndmc.ac.jp; 3Department of Functional Genomics, Chiba University Graduate School of Medicine, Chiba 260-8670, Japan; cada5015@chiba-u.jp (S.A.); naoko-k@hospital.chiba-u.jp (N.K.); 4Department of Otorhinolaryngology/Head and Neck Surgery, Chiba University Graduate School of Medicine, Chiba 260-8670, Japan; thanazawa@faculty.chiba-u.jp

**Keywords:** microRNA, HNSCC, *miR-30e-3p*, tumor suppressor, TCGA, *SERPINE1*

## Abstract

Recently, our studies revealed that some passenger strands of microRNAs (miRNAs) were closely involved in cancer pathogenesis. Analysis of miRNA expression signatures showed that the expression of *miR-30e-3p* (the passenger strand of pre-*miR-30e*) was significantly downregulated in cancer tissues. In this study, we focused on *miR-30e-3p* (the passenger strand of pre-*miR-30e*). We addressed target genes controlled by *miR-30e-3p* that were closely associated with the molecular pathogenesis of head and neck squamous cell carcinoma (HNSCC). Ectopic expression assays demonstrated that the expression of *miR-30e-3p* attenuated cancer cell malignant phenotypes (e.g., cell proliferation, migration, and invasive abilities). Our analysis of *miR-30e-3p* targets revealed that 11 genes (*ADA*, *CPNE8*, *C14orf126*, *ERGIC2*, *HMGA2*, *PLS3*, *PSMD10*, *RALB*, *SERPINE1*, *SFXN1,* and *TMEM87B*) were expressed at high levels in HNSCC patients. Moreover, they significantly predicted the short survival of HNSCC patients based on 5-year overall survival rates (*p* < 0.05) in The Cancer Genome Atlas (TCGA). Among these targets, *SERPINE1* was found to be an independent prognostic factor for patient survival (multivariate Cox regression; hazard ratio = 1.6078, *p* < 0.05). Aberrant expression of *SERPINE1* was observed in HNSCC clinical samples by immunohistochemical analysis. Functional assays by targeting *SERPINE1* expression revealed that the malignant phenotypes (e.g., proliferation, migration, and invasion abilities) of HNSCC cells were suppressed by the silencing of *SERPINE1* expression. Our miRNA-based approach will accelerate our understanding of the molecular pathogenesis of HNSCC.

## 1. Introduction

The Global Cancer Statistics (2018) reported that head and neck squamous cell carcinoma (HNSCC) is the eighth most common human malignancy worldwide [1]. Every year, approximately 800,000 patients are diagnosed with HNSCC, and there are 430,000 deaths [2]. HNSCC is a malignant neoplasm that occurs in the oral cavity, hypopharynx, nasopharynx, and larynx, and the most common subtype is oral squamous cell carcinoma [3]. The aggressive nature of HNSCC is characterized by a high recurrence rate and distant metastasis. Almost 60% of such patients are initially diagnosed at an advanced stage of disease [4]. Treatment strategies for advanced cases are limited and their prognosis is poor, with 5-year survival rates below 30% [5]. The developed immunotherapies are not effective for many patients, and this treatment imposes a financial burden on patients [6]. In order to improve the prognosis of patients, it is essential to better understand the molecular mechanism underlying the malignant features of HNSCC and to develop new therapeutic strategies.

As a result of the human genome project, it became clear that a vast number of non-coding RNA molecules (ncRNAs) are transcribed from the human genome. Those molecules function in both normal and pathological cells [7]. Recent studies have revealed that various ncRNAs play pivotal roles in cell maintenance: for example, fine-tuning gene expression, controlling the cell cycle, and stabilizing RNA molecules, etc. [8,9]. It is apparent that the dysregulation of ncRNAs can contribute to the initiation and enhancement of human diseases, including cancers [10].

Due to their regulatory roles, microRNAs (miRNAs) have been intensively scrutinized. They function as fine-tuners of gene expression in a sequence-dependent manner [11]. A single miRNA might control a great number of RNA transcripts and contribute to various cellular signaling pathways under both normal and pathological conditions [10,12]. A vast number of studies have shown that aberrantly expressed miRNAs can function as oncogenes and/or tumor suppressors in human cancer cells, including HNSCC [13,14].

Two types of mature miRNAs are derived from pre-miRNAs. One strand (the guide strand) is selected for loading into the miRNA-Induced Silencing Complex (miRISC). The miRISC (including the guide strand) targets mRNA for silencing based on sequence compatibility [15]. The other strand of pre-miRNA (the passenger stand) is usually ejected and degraded in the cytoplasm [11,15]. However, recent studies showed that some passenger strands can act as oncogenes or tumor suppressors in a wide range of cancers, targeting several cancer-related genes [16,17,18]. Moreover, recent studies indicated that both strands of pre-miRNAs cooperated to control oncogenic pathways and exerted tumor-suppressive functions in several cancers [19,20,21]. The realization that passenger strands are involved in the molecular pathogenesis of cancer should permit new developments in cancer research.

We have created the miRNA expression signatures in various types of cancers [17,22,23]. Analysis of our miRNA signatures and other studies revealed that some members of the *miR-30* family were frequently downregulated in cancer tissues, suggesting that the *miR-30* family acted as pivotal tumor-suppressors [24,25,26,27]. The Cancer Genome Atlas (TCGA) datasets analysis showed that *miR-30e-3p* (the passenger strand derived from pre-*miR-30e*) was significantly downregulated in HNSCC tissues, and its low expression predicted worse prognosis of the disease. Some genes controlled by *miR-30e-5p* are closely associated with the prognosis of HNSCC patients [28,29]. In contrast, *miR-30e-3p* (the passenger strand) has not been carefully examined in HNSCC.

The aim of this study was to elucidate the involvement of *miR-30e-3p* in HNSCC. Here, ectopic expression assays showed that *miR-30e-3p* had tumor-suppressive roles in HNSCC cells. A total of 53 genes were successfully identified as putative *miR-30e-3p* targets in HNSCC cells. Among these targets, the high expression of 11 genes (*ADA*, *CPNE8*, *C14orf126*, *ERGIC2*, *HMGA2*, *PLS3*, *PSMD10*, *RALB*, *SERPINE1*, *SFXN1,* and *TMEM87B*) significantly predicted the short survival of HNSCC patients according to The Cancer Genome Atlas (TCGA) (5-year overall survival rate; *p* < 0.05). This is the first report of the involvement of *miR-30e-3p* (the passenger strand) and its target genes in HNSCC.

## 2. Results

### 2.1. Expression Levels of miR-30e-3p in Clinical Specimens

The expression levels of *miR-30e-3p* were evaluated by TCGA-HNSC database analysis. The expression levels were significantly downregulated in HNSCC tissues compared with normal tissues (*p* < 0.001; Figure 1A). The downregulation of *miR-30e-3p* in HNSCC tissues was confirmed by other datasets (GSE45238 and GSE31277; Appendix A). The expression of *miR-30e-5p* (the guide strand derived from pre-*miR-30e*) was also downregulated in HNSCC tissues by TCGA database analysis (data not shown). To determine the clinical significance, the 5-year overall survival rates of HNSCC patients were assessed using TCGA-HNSC data. Patients with low expression of *miR-30e-3p* had a significantly poorer prognosis compared to those with high expression [log lank *p* value = 0.0353, hazard ratio (HR) = 0.6097, 95% confidence interval (95% CI): 0.3828–0.9711] (Figure 1B).

### 2.2. Effects of Ectopic Expression of miR-30e-3p on HNSCC Cell Lines

To confirm the antitumor effectiveness of *miR-30e-3p* in HNSCC cells, we conducted ectopic expression assays using two HNSCC cell lines (Sa3 and SAS), focusing on cell proliferation, migration, and invasion. Cancer cell proliferation, migration, and invasion were significantly inhibited by *miR-30e-3p* transfection (Figure 1C–E). Representative images of the migration and invasion assays are shown in Appendix A.

### 2.3. Identification of Putative Oncogenic Targets of miR-30e-3p in HNSCC Cells

To search for the oncogenic targets of HNSCC cells, we combined two datasets to select candidate targets controlled by *miR-30e-3p*. Our strategy for identifying *miR-30e-3p* target genes is shown in Figure 2.

According to the TargetScan database (release 7.2), 6118 genes had putative *miR-30e-3p* binding sites within their sequences (Figure 2). In this study, we performed genome-wide genes expression analysis (oligo-array) using *miR-30e-3p* transfected Sa3 cells, and 241 downregulated genes were identified. Our gene expression data were deposited in the Gene Expression Omnibus (GEO) database (accession number: GSE189290). We selected 150 genes by combining the aforementioned two datasets.

### 2.4. Clinical Significance of miR-30e-3p Target Genes Determined by TCGA Analysis

Clinicopathological analysis of the 150 putative target genes was performed using TCGA data to confirm clinical relevance. Among those 150 genes, 53 were significantly upregulated in HNSCC tissues (*n* = 518) compared with normal tissues (*n* = 44) according to the TCGA-HNSC database (Table 1).

Furthermore, 11 of the genes (*ADA*, *CPNE8*, *C14orf126*, *ERGIC2*, *HMGA2*, *PLS3*, *PSMD10*, *RALB*, *SERPINE1*, *SFXN1,* and *TMEM87B*) predicted a significantly poorer prognosis in HNSCC patients (Figure 3 and Figure 4). To confirm the upregulation of these target genes in HNSCC tissues, we verified using other datasets (GSE30784 and GSE59102; Appendix A). Furthermore, it was examined by quantitative PCR whether these target genes were regulated by *miR-30e-3p* in HNSCC cells. It was revealed that the expression levels of all genes were reduced by the *miR-30e-3p* transfected in HNSCC cells (Appendix A). PCR primer sequences were shown in Appendix A. Among these targets, *SERPINE1* was found to be an independent prognostic factor for patient survival (log rank test, *p* < 0.001 and false discovery rate < 0.05, multivariate Cox regression; hazard ratio = 1.6078, *p* = 0.0031; Table 1, Figure 5A).

Moreover, we investigated the extent to which the expression of *miR-30e-3p* and *SERPINE1* was correlated in HNSCC clinical specimens. Spearman’s rank test revealed a negative correlation between the expression levels of *miR-30e-3p* and *SERPINE1* (*p* < 0.01, *r* = −0.3717; Figure 5B).

Immunohistochemistry was performed to analyze SERPINE1 protein expression in HNSCC clinical specimens. *SERPINE1* protein was clearly stained in cancer lesions, whereas it was only weakly stained in normal tissues (Figure 5C).

### 2.5. Regulation of SERPINE1 Expression by miR-30e-3p in HNSCC Cells

Both the mRNA and protein expression levels of *SERPINE1* were suppressed in *miR-30e-3p*-transfected HNSCC cells (Figure 6A,B). Full-size Western blot images are shown in Appendix A. To investigate whether *miR-30e-3p* bound directly to the 3′-UTR of *SERPINE1* in HNSCC cells, dual-luciferase reporter assays were performed. TargetScan database analysis revealed that there were two *miR-30e-3p* binding sites predicted within the 3′-UTR of *SERPINE1* (Figure 6C). Luciferase activity was significantly reduced following the co-transfection of *miR-30e-3p* and a vector containing the *miR-30e-3p*-binding site (1322 to 1328) in the 3′-UTR of *SERPINE1*. On the other hand, co-transfection of a vector containing the *SERPINE1* 3′-UTR lacking the *miR-30e-3p*-binding site resulted in no change in luciferase activity (left panel of Figure 6C). With regard to the other predicted binding site (1607 to 1614), luciferase activity was not changed following co-transfection of *miR-30e-3p* and a vector containing the *miR-30e-3p*-binding site in the 3′-UTR of *SERPINE1* (right panel of Figure 6C). Our present data suggest that *miR-30e-3p* binds directly to one of the predicted binding sites of *SERPINE1*, and that it controls the expression levels of *SERPINE1* in HNSCC cells.

Dual luciferase reporter assays showed reduced luminescence activity after co-transfection of the wild-type vector (position 1322–1328) and *miR-30e-3p* in Sa3 cells (left panel). There was no reduced luminescence activity after co-transfection of the deletion-type vector (position 1322–1328) and *miR-30e-3p* in Sa3 cells (lower panel). Normalized data are expressed as the Renilla/firefly luciferase activity ratio (N.S., not significant). For the other putative binding site (position 1607–1614), there was no reduced luminescence activity after co-transfection of the wild-type vector and *miR-30e-3p* in Sa3 cells (right panel).

### 2.6. Effects of SERPINE1 Knockdown on the Proliferation, Migration, and Invasion Assays

To explore the potential cancer-promoting function of *SERPINE1* in HNSCC cells, we used siRNAs targeting *SERPINE1* in knockdown assays. First, the inhibitory effects of two different siRNAs targeting *SERPINE1* (si*SERPINE1–*1 and si*SERPINE1–*2) on *SERPINE1* expression were confirmed. The *SERPINE1* mRNA and protein levels were effectively suppressed by each siRNA transfected into Sa3 and SAS cells (Appendix A). Knockdown of *SERPINE1* in HNSCC cells inhibited cancer cell malignant transformation (Figure 7A–C). Notably, cancer cell invasion and migration abilities were significantly blocked after si*SERPINE1–*1 or si*SERPINE1–*2 was transfected into Sa3 and SAS cells (Figure 7B,C). Our present data indicate that aberrantly expressed *SERPINE1* contributed to the aggressive phenotype of HNSCC cells.

To understand the effects of overexpression of *SERPINE1* in HNSCC cells, gene set enrichment analysis (GSEA) was performed to determine differentially expressed genes between the high and low *SERPINE1* expression groups of the TCGA-HNSC cohort. The results of GSEA showed that the most enriched gene set in the high *SERPINE1* expression group was “epithelial–mesenchymal transition” (Figure 7D and Table 2). GSEA analysis showed that several signal pathways were involved in the high *SERPINE1* expression group, e.g., “myogenesis”, “TNFα signaling”, “angiogenesis”, and “KRAS signaling” (Table 2). We suggest that the overexpression of *SERPINE1* may affect various intracellular signaling pathways. Activation of these signaling pathways may induce the malignant transformation of HNSCC cells.

## 3. Discussion

Initially, it was believed that only the guide strand derived from pre-miRNA was functional, whereas the passenger strand was degraded and had no function [11,15]. Contrary to this concept, a large number of studies have shown that the passenger strands of several miRNAs are responsible for the regulation of target genes in the cell [30]. A large number of cohort analyses by TCGA data showed that in some miRNAs (e.g., *miR-30a*, *miR-143*, *miR-145*, and *miR-139*), both strands (-5p/-3p (the guide strand/the passenger strand)) cooperated to regulate oncogenic pathways [30]. Our recent studies demonstrated that *miR-30a-3p*, *miR-143-5p*, *miR-145-3p,* and *miR-139-3p* (all of which are passenger strands) acted as tumor-suppressive miRNAs through the control of many oncogenic genes and pathways in various types of cancers [31,32,33]. Finding functional passenger chains and exploring the networks they control will deepen our understanding of the molecular mechanisms of cancer.

The ectopic expression of *miR-30e-5p* markedly blocked the malignant characteristics of certain cancer cells, indicating that *miR-30e-5p* acted as a tumor-suppressor in HNSCC cells [28]. Our present analysis revealed that the other strand derived from pre-*miR-30e* also has a tumor-suppressive function. In other words, two miRNAs (*miR-30e-5p* and *miR-30e-3p*) derived from pre-*miR-30e* function as tumor suppressors in HNSCC cells.

Relatively little is known about the functional characteristics of *miR-30e-3p* in human cancers. A previous study showed that the overexpression of *miR-30e-3p* inhibited renal cell carcinoma (RCC) cell line (A498 and 786-O) migration and invasive abilities [34]. This study also indicated that *Snail1* was directly regulated by *miR-30e-3p* in RCC cells [34]. *Snail1* is a zinc finger-containing transcription factor. It functions as a negative regulator of E-cadherin expression, and it is closely related to RCC metastasis [35].

Recently, an interesting study of *miR-30e-3p* reported that *miR-30e-3p* possessed two functions (tumor-suppressor or oncogene) depending on *TP53* status [36]. With wild-type TP53, *miR-30e-3p* targeted *MDM2*, and it seems to behave as a tumor suppressor. In contrast, with a nonfunctional *TP53*, *miR-30e-3p* behaved as an oncogene. Notably, elevated *miR-30e-3p* levels predicted the development of sorafenib resistance in hepatocellular carcinoma patients [36]. It is an interesting finding that the role of *miR-30e-3p* varies depending on the status of the *TP53*. It is necessary to examine whether this situation is a universal phenomenon in HNSCC cells.

A single miRNA can control a large number of genes, and the target genes can differ between cell types. Therefore, it is important to clarify which genes are controlled by *miR-30e-3p* for each type of cancer. Our in silico analysis of *miR-30e-3p* targets revealed that 53 genes acted as putative oncogenic targets in HNSCC. The expression of all 11 genes was closely associated with the molecular pathogenesis of HNSCC. A detailed analysis of these genes should improve our understanding of the molecular mechanisms underlying HNSCC. For example, Plastin-3 (*PLS3*) is an actin-bundling protein that contributes to cofilin-mediated actin polymerization [37]. The aberrant expression of PLS3 was reported in a wide range of cancers, and its expression is closely associated with the EMT-induced malignant phenotypes of cancers [38]. A previous study showed that the expression of *PLS3* by circulating tumor cells was a marker in metastatic colorectal cancer [39]. Recent study indicated that it is possible to predict the responsiveness of lung cancer patients to nivolumab treatment by measuring plasma levels of PLS3 [40].

PSMD10 (Proteasome 26S Subunit, Non-ATPase 10), also known as Gankyrin, was initially cloned from a cDNA library obtained from a hepatocellular carcinoma. It consisted of ankyrin repeat motifs that promoted protein–protein interactions [41]. A large number of studies showed that aberrantly expressed PSMD10/Gankyrin was present in several types of cancers, and its overexpression enhanced cancer cell proliferation, invasion, and metastasis through the activation of several oncogenic signal pathways, e.g., RhoA/ROCK/PTEN, PI3K/AKT, and IL-6/STAT3 pathways [42]. In oral cancer, overexpressed PSMD10/Gankyrin was detected in cancer tissues and premalignant oral lesions [43].

Human RAL (RAS-like) is a member of the RAS-family (RAS, RAL, RIT, RAP, RHEB, and RAD), and it has an amino acid sequence most similar to RAS [44]. The human *RALA* and *RALB* genes were cloned from a human pheochromocytoma cDNA library. The two proteins share approximately 80% similarity at the amino acid level [45]. Previous studies showed that RALA and RALB might play distinctive oncogenic roles in human cancers, as RALB contributed to the survival of human cancer cells [46]. In pancreatic cancer, RALB was activated in cancer tissues, and it enhanced the cells’ invasive ability and their metastatic colonization in in vitro and in vivo studies [47].

*SERPINE1* (also known as Plasminogen Activator Inhibitor-1: *PAI-1*) is an inhibitor of the plasminogen activators tPA and uPA [48]. In a wide range of cancers, the upregulation of *SERPINE1* was observed in genome-wide gene expression analyses, and its expression is a marker of poor prognosis [49]. The roles of *SERPINE1* in cancer progression have been studied in depth, especially tumor promotion of inflammation, sustaining proliferative signals, invasion and metastasis, angiogenesis, and resisting tumor death [50]. In genome-wide gene expression profiles of HNSCC, upregulated *SERPINE1* was frequently observed in independent studies [51]. A previous study showed that the overexpression of *SERPINE1* enhanced the migration of HNSCC cells, and its expression protected cells from cisplatin-induced apoptosis through activation of the PI3K/AKT pathway [52]. Another study indicated that overexpression of *SERPINE1* also promoted tumor aggressiveness and metastatic dissemination to lymph nodes and lung. Moreover, its association is consistent with poor outcome in HNSCC patients [53]. Given its predominance in the HNSCC literature, it is surprising that no effective cancer chemotherapy targeting *SERPINE1* has been proposed.

Accumulating studies have shown that miRNAs regulated the malignancy of cancer cells by controlling the target genes [54]. The downregulation of tumor-suppressive miRNAs causes the aberrant expression of oncogenic genes in cancer cells. The involvement of several tumor-suppressive microRNAs has been reported as a caused of overexpression of *SERPINE1* in cancer cells [52]. Previous study showed that *miR-617* was directly bound in the UTR of *SERPINE1* mRNA and controlled its expression in OSCC cells [55]. Other study demonstrated that ectopic expression of *miR-181c-5p* suppressed the expression of *SERPINE1* in HNSCC cells [56]. Notably, the expression of *miR-617* and *miR-181c-5p* inhibited cancer cell proliferation and migration abilities in OSCC/HNSCC cells [55,56].

Here, we identified a number of genes that were controlled by *miR-30e-3p.* Many of those molecules are involved in cancer pathogenesis. Functional analysis of these genes will reveal at least some of the molecular mechanisms underlying HNSCC.

## 4. Materials and Methods

### 4.1. Transfection of Mature miRNAs and Small-Interfering RNAs (siRNAs) into HNSCC Cells

The HNSCC cell lines (SAS and Sa3) used in this study were obtained from the RIKEN BioResource Center (Tsukuba, Ibaraki, Japan). The miRNA precursors, negative control miRNA, and siRNAs were obtained from Applied Biosystems (Waltham, MA, USA). The procedures used for the transient transfection of miRNAs, siRNAs, and plasmid vectors were described in our previous studies [20,21,57,58]. miRNAs at 10 nM and siRNAs at 5 nM were transfected into HNSCC cell lines using RNAiMAX reagent (Invitrogen, Waltham, MA, USA). The reagents used in this study are listed in Appendix A; here, “mock”: transfection reagent only, and “control”: negative control miRNA precursor that have no function transfected.

### 4.2. Functional Assays of miR-30e-3p and SERPINE1 on HNSCC Cells

Cell proliferation, migration, and invasion assays were performed using HNSCC cells. The XTT assay (Sigma–Aldrich, St. Louis, MO, USA) characterized cell proliferation, a chamber assay using the Corning BioCoat^TM^ cell culture chamber (Corning, Corning, NY, USA) assessed migration, and Matrigel chamber assays using the Corning BioCoat Matrigel assessed invasion. They were performed with HNSCC cells as described previously [20,21,57,58]. The reagents used in this study are listed in Appendix A.

### 4.3. RNA Extraction and Quantitative Real-Time Reverse Transcription Polymerase Chain Reaction (qRT-PCR)

RNA extraction from cell lines and qRT-PCR were performed as described in our previous studies [20,21,57,58]. The TaqMan assays used in this study are summarized in Appendix A. The primers for SYBR green assays designed in this study are summarized in Appendix A. *GAPDH* was used as the internal control.

### 4.4. Western Blotting and Immunohistochemistry

Western blotting and immunohistochemical procedures were described in our previous studies [20,21,48,49]. Paraffin sections for immunohistochemistry were obtained from HNSCC patients who underwent surgical treatment at Chiba University hospital. The clinical features of the HNSCC patients are summarized in Appendix A. The antibodies used in this study are shown in Appendix A. Full blots of the membrane are shown in Appendix A. Our study has been approved by the Ethics Committee of Chiba University (approval number; 28–65, 10 February 2015). The research methodology is implemented in accordance with the standards set by the Declaration of Helsinki.

### 4.5. Dual Luciferase Reporter Assays

Synthetic DNA sequences of *SERPINE1* with or without the *miR-30e-3p*-binding sites were inserted into the psiCHECK-2 vector (C8021; Promega, Madison, WI, USA). The differences between wild-type and deletion type sequences are detailed in Figure 6. Then, the plasmid vectors and *miR-30e-3p* were transfected into Sa3 cells. After 48 h of transfection, dual luciferase reporter assays were performed using the Dual Luciferase Reporter Assay System (Promega). Normalized data are expressed as a Renilla/Firefly luciferase activity ratio.

### 4.6. Identification of miR-30e-3p Targets in HNSCC

To predict miRNA binding sites, we relied on the TargetScanHuman database (release 7.2, http://www.targetscan.org/vert_72/; accessed on 1 August 2019) [59]. We assessed the genes that were downregulated by the transfection of *miR-30e-3p into* HNSCC cells using genome-wide gene expression analysis (microarray assays). Our expression data were deposited in the GEO database (accession number: GSE189290). The strategy used for the identification of *miR-30e-3p* target genes in this study is summarized in Figure 2.

To validate the expression of *miR-30e-3p* and identified target genes in multiple HNSCC samples, GSE45238, GSE31277, GSE30784, and GSE59102 were downloaded from GEO datasets.

### 4.7. Analysis of miRNAs and miRNA Target Genes in HNSCC Patients

We used TCGA-HNSC (https://tcga-data.nci.nih.gov/tcga/; accessed on 10 July 2019) to investigate the clinical significance of miRNAs and their target genes. Clinical parameters and gene expression data were obtained from cBioPortal (http://www.cbioportal.org/; accessed on 13 July 2019) [60,61] and OncoLnc (http://www.oncolnc.org/; accessed on 1 August 2019) [62].

For Kaplan–Meier analyses of overall survival, we downloaded TCGA clinical data (Firehose Legacy) from cBioPortal (https://www.cbioportal.org; data downloaded on 13 July 2019). Gene expression grouping data for each gene were collected from OncoLnc (http://www.oncolnc.org/; accessed on 1 August 2019).

Monovariate and multivariate analyses were also performed using TCGA-HNSC clinical data and survival data according to the expression level of each gene from OncoLnc to identify factors associated with HNSCC patient survival. In addition to gene expression, the tumor stage, pathological grade, and age were evaluated as potential independent prognostic factors. The multivariate analyses were performed using JMP Pro 15.0.0 (SAS Institute Inc., Cary, NC, USA).

### 4.8. Gene Set Enrichment Analysis (GSEA)

To analyze the molecular pathways related to *SERPINE1*, we performed GSEA. We divided TCGA-HNSC data into high and low expression groups according to the Z-score of the *SERPINE1* expression level. A ranked list of genes was generated by log2 ratio comparing the expression levels of each gene between the two groups. The obtained gene lists were uploaded into GSEA software [63,64]. The Hallmark gene set in The Molecular Signatures Database was applied [63,65].

### 4.9. Statistical Analysis

All data are shown as mean values with standard errors derived from ≥3 independent experiments. Statistical analyses were determined using JMP Pro 15 (SAS Institute Inc., Cary, NC, USA). Welch’s *t*-test was performed to determine the significance of differences between two groups. Dunnet’s test was applied for comparisons among multiple groups. We applied Spearman’s rank test to evaluate the correlation between expression of *miR-30e-3p* and target genes. Monovariate and multivariate analyses were performed using Cox’s proportional hazards model. A *p*-value less than 0.05 was considered statistically significant.

## 5. Conclusions

In conclusion, analysis of the TCGA database showed that the expression level of *miR-30e-3p* was significantly downregulated in HNSCC clinical specimens. Ectopic expression assays demonstrated that *miR-30e-3p* inhibited HNSCC cells’ aggressiveness, suggesting that *miR-30e-3p* acted as a tumor suppressor. Interestingly, some of the putative targets regulated by *miR-30e-3p* closely predicted the prognosis of HNSCC patients. Our identification of *miR-30e-3p* (the passenger strand of pre-*miR-30e*) in the molecular pathogenesis of HNSCC may open the way to improved treatments of HNSCC oncogenesis.

## Figures and Tables

**Figure 1 ijms-23-03808-f001:**
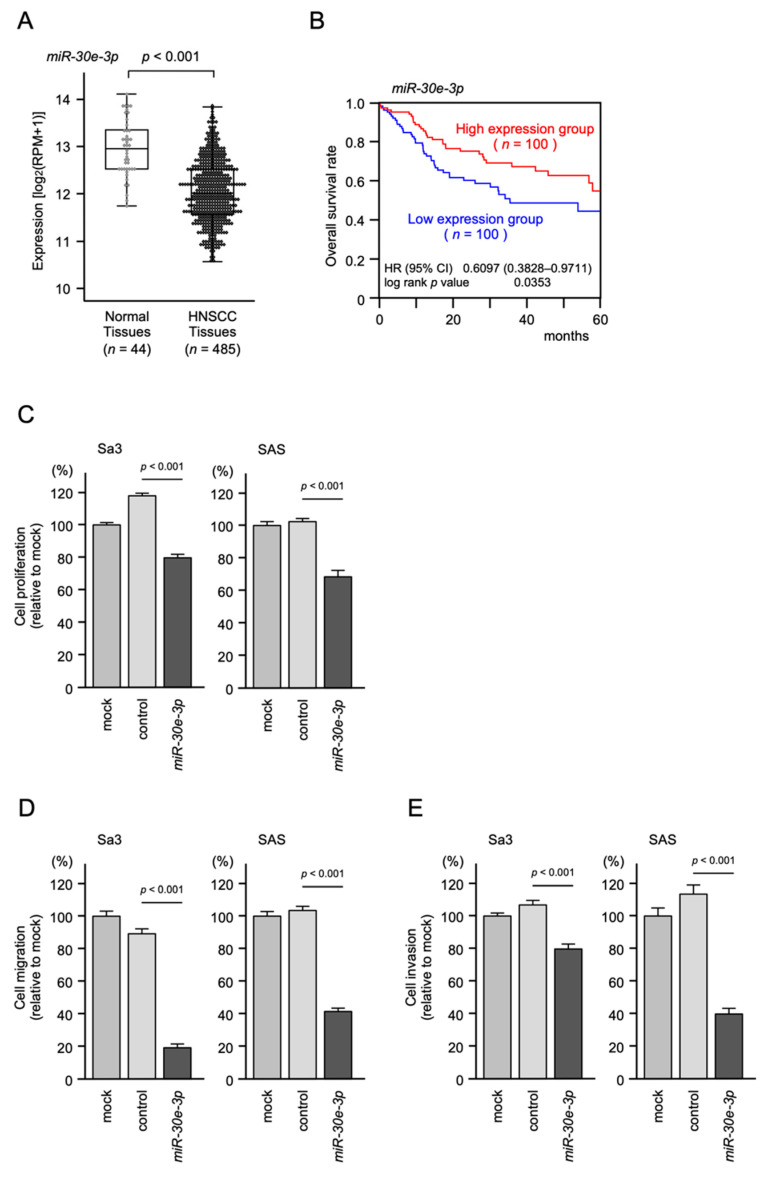
Tumor-suppressive role of *miR-30e-3p* in HNSCC cells. (**A**) The expression level of *miR-30e-3p* was analyzed using the TCGA-HNSC database. A total of 485 HNSCC tissues and 44 normal epithelial tissues were evaluated. (**B**) Kaplan–Meier survival analyses of HNSCC patients using data from the TCGA database. Patients were divided into two groups (top 25% and low 25%). The red and blue lines indicate the high and low expression groups, respectively (log rank *p* value = 0.0353, HR = 0.6097, 95% CI: 0.3828–0.9711). (**C**–**E**) Functional assays of cell proliferation, migration, and invasion following the transient transfection of *miR-30e-3p* in HNSCC cell lines (Sa3 and SAS cells). (**C**) Cell proliferation assessed by XTT assay at 72 h after siRNA transfection. (**D**) Cell migration assessed using a membrane culture system at 48 h after seeding miRNA-transfected cells into the chambers. (**E**) Cell invasion assessed by Matrigel invasion assays at 48 h after seeding miRNA-transfected cells into chambers.

**Figure 2 ijms-23-03808-f002:**
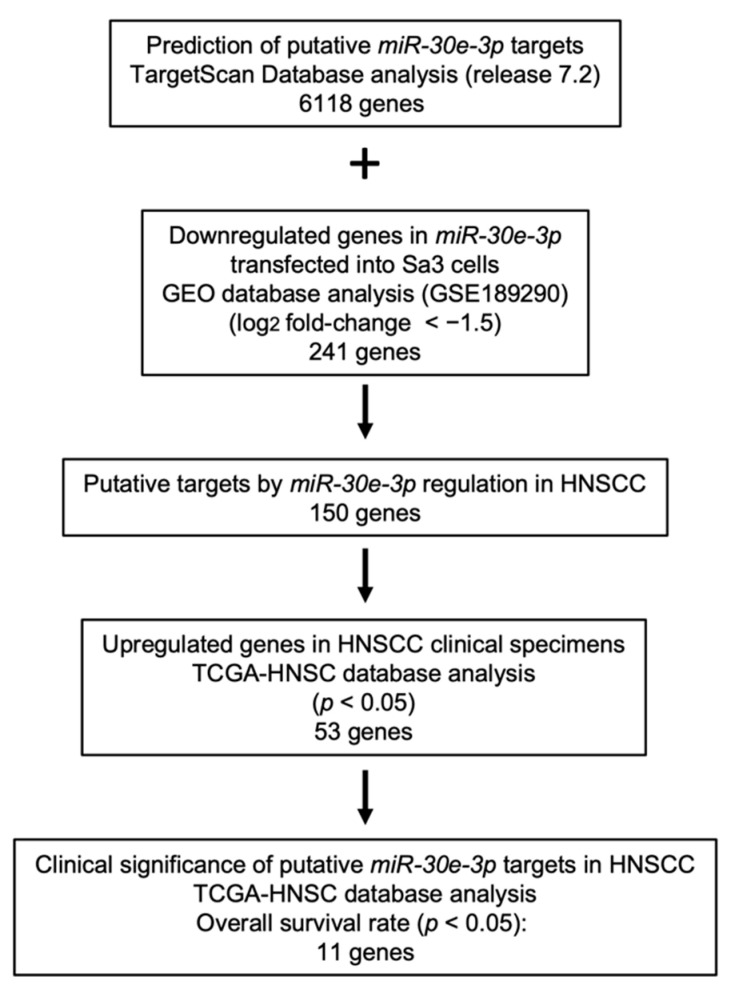
Flowchart of the strategy used to identify candidate *miR-30e-3p* target genes in HNSCC cells.

**Figure 3 ijms-23-03808-f003:**
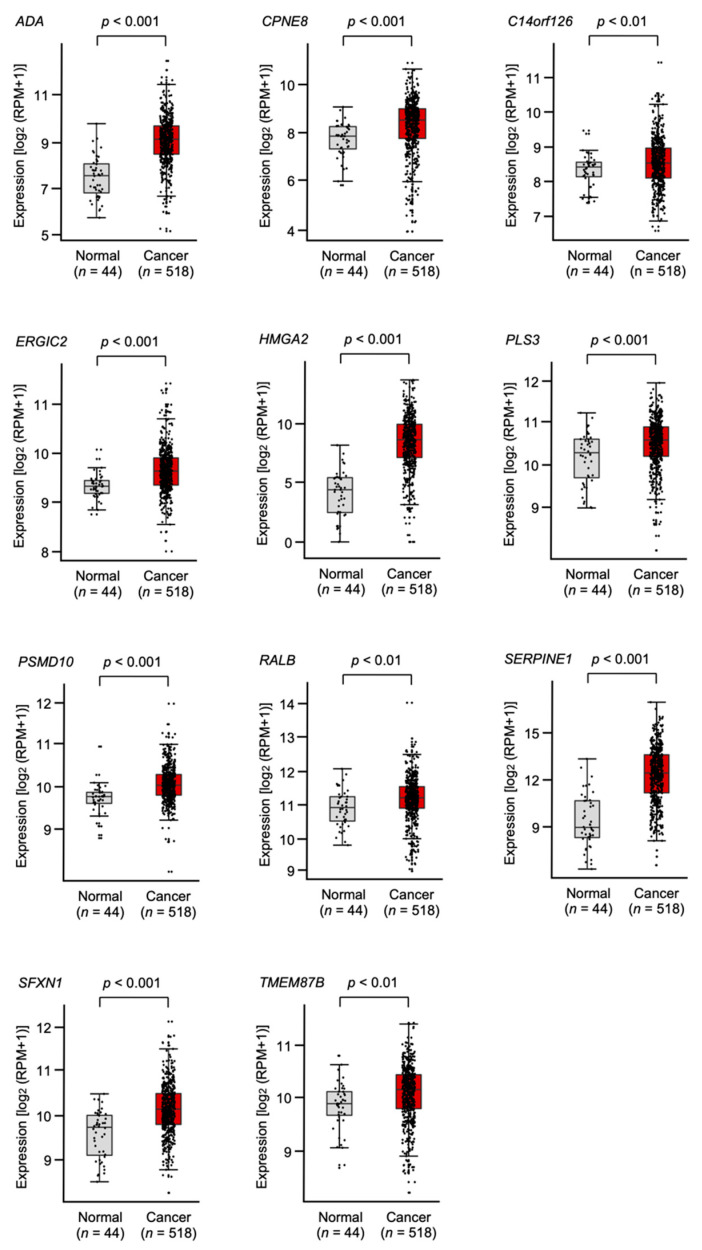
HNSCC tissue expression of 11 target genes of *miR-30e-3p* using TCGA-HNSC data. The expression levels of 11 genes (*ADA*, *CPNE8*, *C14orf126*, *ERGIC2*, *HMGA2*, *PLS3*, *PSMD10*, *RALB*, *SERPINE1*, *SFXN1*, and *TMEM87B*) were analyzed using the TCGA-HNSC database. A total of 518 HNSCC tissues and 44 normal epithelial tissues were evaluated.

**Figure 4 ijms-23-03808-f004:**
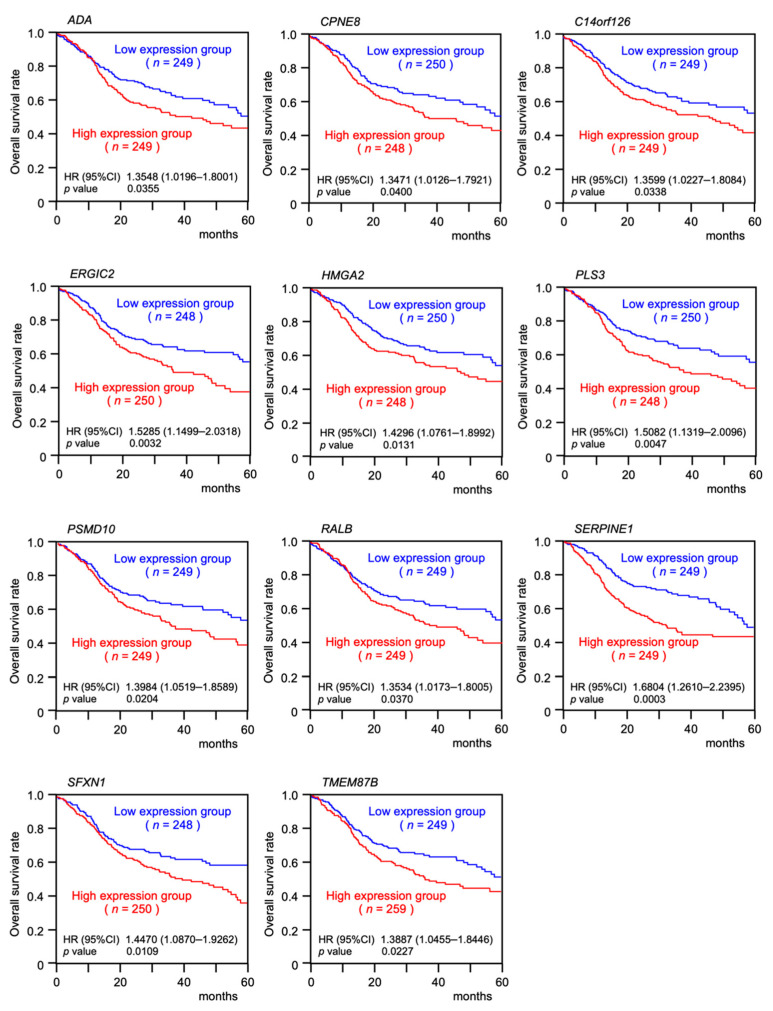
Five-year survival rates of 11 target genes of *miR-30e-3p* using TCGA-HNSC data. Kaplan–Meier survival analyses of HNSCC patients using data from the TCGA database. Patients were divided into high and low expression groups according to the median of each gene expression level. The red and blue lines indicate the high and low expression groups, respectively.

**Figure 5 ijms-23-03808-f005:**
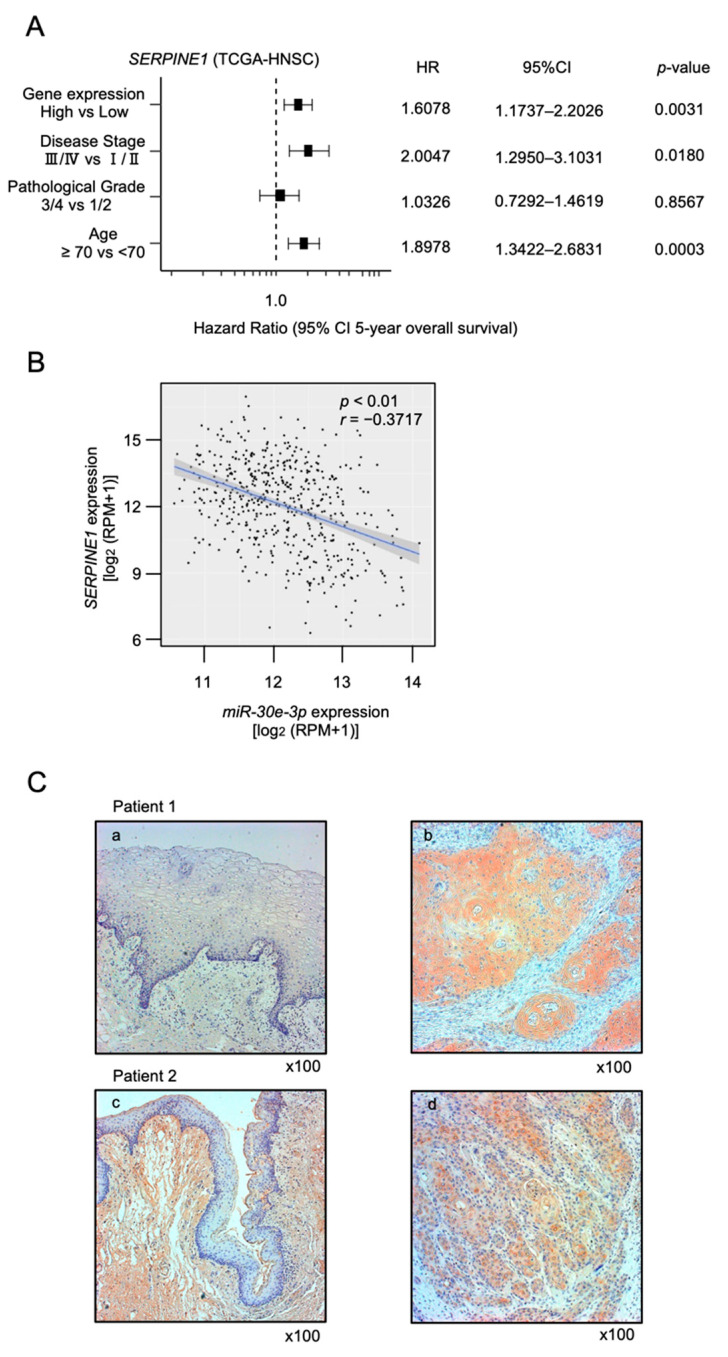
Clinical significance of *SERPINE1* using TCGA-HSCC data. (**A**) Forest plot showing the multivariate analysis results for *SERPINE1*, which were identified as independent prognostic factors for overall survival after adjustment for patient age, disease stage, and pathological grade. (**B**) Expression negative correlation between *miR-30e-3p* and *SERPINE1* in HNSCC clinical specimens. Spearman’s rank test indicated negative correlations of *miR-30e-3p* expression with *SERPINE1* (*p* < 0.01, *r* = −0.3717). (**C**) Immunohistochemical staining of *SERPINE1* in HNSCC clinical specimens. *SERPINE1* expression was high in the cancer lesions (right panels; b and d), whereas normal mucosa were only weakly stained (left panels; a and c).

**Figure 6 ijms-23-03808-f006:**
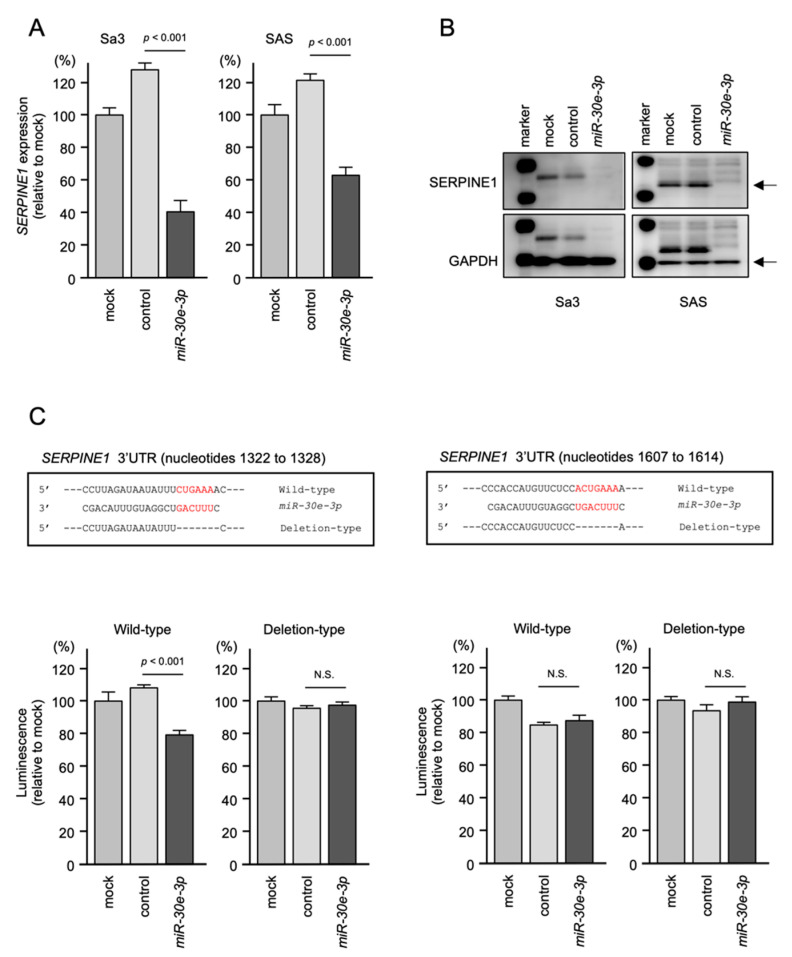
Direct regulation of *SERPINE1* expression by *miR-30e-3p* in HNSCC cells. (**A**) qRT-PCR showing significantly reduced expression of *SERPINE1* mRNA at 72 h after *miR-30e-3p* transfection in Sa3 and SAS cells. Expression of *GAPDH* was used as an internal control. (**B**) Western blot showing reduced expression of *SERPINE1* protein at 72 h after *miR-30e-3p* transfection in Sa3 and SAS cells. Expression of GAPDH was used as an internal control. (**C**) TargetScan database analysis predicting two putative *miR-30e-3p*-binding sites in the 3′-UTR of *SERPINE1* (upper panel).

**Figure 7 ijms-23-03808-f007:**
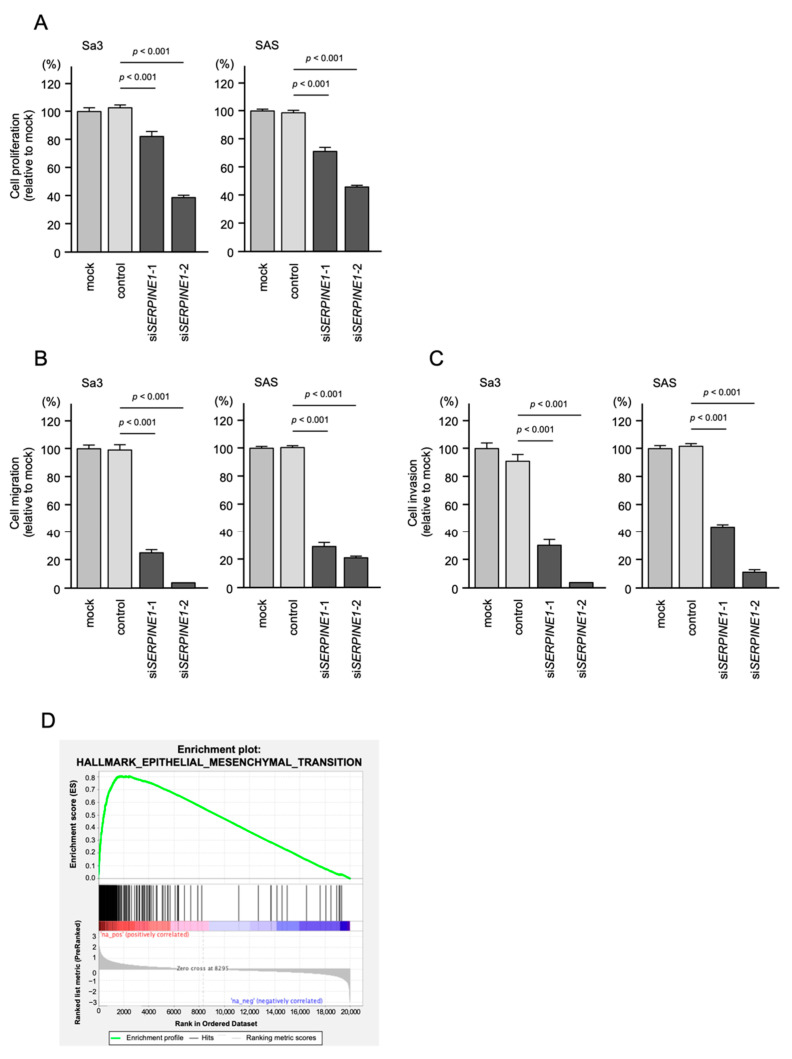
Functional assays of cell proliferation, migration, and invasion following the transient transfection of siRNAs targeting *SERPINE1* in HNSCC cell lines (Sa3 and SAS cells). (**A**–**C**) Functional assays of cell proliferation, migration, and invasion following the transient transfection of si*SERPINE1–*1 *and* si*SERPINE1–*2 in HNSCC cell lines (Sa3 and SAS cells). (**A**) Cell proliferation assessed by XTT assay at 72 h after siRNA transfection. (**B**) Cell migration assessed using a membrane culture system at 48 h after seeding miRNA-transfected cells into the chambers. (**C**) Cell invasion assessed by Matrigel invasion assays at 48 h after seeding miRNA-transfected cells into chambers. (**D**) GSEA analysis showed that most enrichment pathway was “epithelial–mesenchymal transition”.

**Table 1 ijms-23-03808-t001:** Upregulated genes in HNSCC clinical specimens in TCGA-HNSC database analysis.

Entrez Gene ID	Gene Symbol	Gene Name	Total Binding Sites	GEO log_2_ FC ^1^	5y OS ^2^ *p*-Value	FDR ^3^
5054	*SERPINE1*	serpin peptidase inhibitor, clade E (nexin, plasminogen activator inhibitor type 1), member 1	2	−1.98	0.0003	0.0209
51290	*ERGIC2*	ERGIC and golgi 2	2	−1.68	0.0032	0.0730
5358	*PLS3*	plastin 3	2	−1.53	0.0047	0.0898
94081	*SFXN1*	sideroflexin 1	2	−1.92	0.0109	0.1423
8091	*HMGA2*	high mobility group AT-hook 2	1	−2.65	0.0131	0.1575
5716	*PSMD10*	proteasome (prosome, macropain) 26S subunit, non-ATPase, 10	1	−1.52	0.0204	0.2014
84910	*TMEM87B*	transmembrane protein 87B	1	−1.64	0.0227	0.2137
112487	*C14orf126*	D-tyrosyl-tRNA deacylase 2 (putative)	1	−1.58	0.0338	0.2663
100	*ADA*	adenosine deaminase	1	−1.64	0.0355	0.2736
5899	*RALB*	v-ral simian leukemia viral oncogene homolog B	2	−1.56	0.0370	0.2799
144402	*CPNE8*	copine VIII	1	−1.74	0.0400	0.2921
10923	*SUB1*	SUB1 homolog (S. cerevisiae)	3	−2.53	0.0933	0.4584
6566	*SLC16A1*	solute carrier family 16 (monocarboxylate transporter), member 1	4	−1.75	0.0955	0.4640
3336	*HSPE1*	heat shock 10kDa protein 1 (chaperonin 10)	1	−1.53	0.0960	0.4652
55156	*ARMC1*	armadillo repeat containing 1	4	−1.91	0.0997	0.4743
79624	*C6orf211*	chromosome 6 open reading frame 211	4	−1.50	0.1251	0.5316
528	*ATP6V1C1*	ATPase, H+ transporting, lysosomal 42kDa, V1 subunit C1	2	−2.04	0.1309	0.5436
64841	*GNPNAT1*	glucosamine-phosphate N-acetyltransferase 1	3	−1.73	0.1564	0.5927
10552	*ARPC1A*	actin related protein 2/3 complex, subunit 1A, 41 kDa	1	−1.97	0.1871	0.6449
6780	*STAU1*	staufen double-stranded RNA binding protein 1	3	−1.77	0.1883	0.6468
9265	*CYTH3*	cytohesin 3	2	−1.52	0.2113	0.6817
90874	*ZNF697*	zinc finger protein 697	3	−1.58	0.2182	0.6915
5923	*RASGRF1*	Ras protein-specific guanine nucleotide-releasing factor 1	2	−1.51	0.2359	0.7157
136	*ADORA2B*	adenosine A2b receptor	1	−2.09	0.2367	0.7168
81539	*SLC38A1*	solute carrier family 38, member 1	5	−2.00	0.2493	0.7330
10473	*HMGN4*	high-mobility group nucleosomal binding domain 4	2	−1.84	0.2678	0.7555
51762	*RAB8B*	RAB8B, member RAS oncogene family	4	−1.59	0.2683	0.7561
54165	*DCUN1D1*	DCN1, defective in cullin neddylation 1, domain containing 1	5	−1.52	0.3327	0.8236
84056	*KATNAL1*	katanin p60 subunit A-like 1	2	−1.55	0.3663	0.8530
3556	*IL1RAP*	interleukin 1 receptor accessory protein	1	−3.20	0.3916	0.8729
112399	*EGLN3*	egl-9 family hypoxia-inducible factor 3	1	−2.10	0.4389	0.9051
1021	*CDK6*	cyclin-dependent kinase 6	3	−1.82	0.4733	0.9249
54108	*CHRAC1*	chromatin accessibility complex 1	1	−1.65	0.4733	0.9249
7172	*TPMT*	thiopurine S-methyltransferase	1	−1.66	0.4754	0.9260
2113	*ETS1*	v-ets avian erythroblastosis virus E26 oncogene homolog 1	3	−1.89	0.5119	0.9436
51199	*NIN*	ninein (GSK3B interacting protein)	2	−1.58	0.5582	0.9616
8862	*APLN*	apelin	1	−2.05	0.5596	0.9621
5597	*MAPK6*	mitogen-activated protein kinase 6	2	−2.10	0.5818	0.9690
57045	*TWSG1*	twisted gastrulation BMP signaling modulator 1	1	−1.69	0.5857	0.9701
55142	*HAUS2*	HAUS augmin-like complex, subunit 2	4	−1.98	0.5862	0.9702
4015	*LOX*	lysyl oxidase	1	−1.55	0.6133	0.9769
4678	*NASP*	nuclear autoantigenic sperm protein (histone-binding)	2	−1.74	0.6811	0.9864
55824	*PAG1*	phosphoprotein associated with glycosphingolipid microdomains 1	5	−2.10	0.7043	0.9871
3553	*IL1B*	interleukin 1, beta	1	−2.15	0.7316	0.9871
65062	*ALS2CR4*	transmembrane protein 237	1	−1.86	0.7378	0.9871
51715	*RAB23*	RAB23, member RAS oncogene family	2	−1.70	0.7820	0.9871
4603	*MYBL1*	v-myb avian myeloblastosis viral oncogene homolog-like 1	1	−1.55	0.7897	0.9871
4893	*NRAS*	neuroblastoma RAS viral (v-ras) oncogene homolog	3	−2.53	0.7935	0.9871
84668	*FAM126A*	family with sequence similarity 126, member A	2	−1.61	0.8471	0.9871
51633	*OTUD6B*	OTU domain containing 6B	2	−1.63	0.9276	0.9871
3930	*LBR*	lamin B receptor	1	−2.40	0.9492	0.9871
8869	*ST3GAL5*	ST3 beta-galactoside alpha-2,3-sialyltransferase 5	1	−2.38	0.9686	0.9871
1719	*DHFR*	dihydrofolate reductase	4	−1.96	0.9722	0.9871

^1^ Fold Change, ^2^ 5-Year Overall Survival, ^3^ False Discovery Rate.

**Table 2 ijms-23-03808-t002:** The significantly enriched gene sets in the high *SERPINE1* expression group in TCGA-HNSC.

Name	Normalized Enrichment Score	FDR *q*-Value
HALLMARK_EPITHELIAL_MESENCHYMAL_TRANSITION	3.149	*q* < 0.001
HALLMARK_MYOGENESIS	2.866	*q* < 0.001
HALLMARK_TNFA_SIGNALING_VIA_NFKB	2.489	*q* < 0.001
HALLMARK_ANGIOGENESIS	2.349	*q* < 0.001
HALLMARK_KRAS_SIGNALING_UP	2.289	*q* < 0.001
HALLMARK_COAGULATION	2.285	*q* < 0.001
HALLMARK_HYPOXIA	2.282	*q* < 0.001
HALLMARK_APICAL_JUNCTION	2.247	*q* < 0.001
HALLMARK_UV_RESPONSE_DN	2.189	*q* < 0.001
HALLMARK_INFLAMMATORY_RESPONSE	2.156	*q* < 0.001
HALLMARK_INTERFERON_ALPHA_RESPONSE	2.054	*q* < 0.001
HALLMARK_TGF_BETA_SIGNALING	2.027	*q* < 0.001
HALLMARK_INTERFERON_GAMMA_RESPONSE	1.896	*q* < 0.001
HALLMARK_COMPLEMENT	1.846	*q* < 0.001
HALLMARK_IL6_JAK_STAT3_SIGNALING	1.836	*q* < 0.001
HALLMARK_NOTCH_SIGNALING	1.793	0.001
HALLMARK_APOPTOSIS	1.716	0.002
HALLMARK_HEDGEHOG_SIGNALING	1.692	0.002
HALLMARK_IL2_STAT5_SIGNALING	1.618	0.004
HALLMARK_GLYCOLYSIS	1.593	0.005
HALLMARK_P53_PATHWAY	1.396	0.038
HALLMARK_WNT_BETA_CATENIN_SIGNALING	1.376	0.043

## Data Availability

Our expression data were deposited in the GEO database (accession number: GSE189290).

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
