# Peer review of "Identification of Tumor-Suppressive miR-30e-3p Targets: Involvement of SERPINE1 in the Molecular Pathogenesis of Head and Neck Squamous Cell Carcinoma"

_ijms, 2022, doi:10.3390/ijms23073808_

Round 1
Reviewer 1 Report
The paper entitled “Identification of tumor-suppressive miR-30e-3p targets: Involvement of SERPINE1 in the molecular pathogenesis of head and neck squamous cell carcinoma” by Minemura C et al., have identified a potential tumor suppressive role of miR-30e-3p in HNSCC and its putative target SERPINE1 as an independent prognostic factor for patient survival. This is an interesting study; however, there are several concerns regarding the manuscript, please find them below:
Major comments
The main concern of this reviewer is novelty. There are reports of miRNAs (for example, miR-617, miR-181c-5p) targeting SERPINE1 in two of the most common forms of HNSCC, oral squamous cell carcinoma (Zhao and Liu, mol cell biol, 2021) and Laryngeal squamous cell carcinoma (Li X, Front. Oncol., 2020), respectively. Additionally, SERPINE1 is a well-established oncogene in several cancers including HNSCC (Pavon MA, oncotarget, 2015).
One major experiment Minemura C et al., have done is the downregulation of SERPINE1 expression in HNSCC cell which the first two papers have not done, still according to this reviewer, the manuscript in its current format is not a substantial advancement in the field of HNSCC.
Apart from the above, there are several other comments/concerns that needs to be addressed.
- Validation of the TCGA analysis for miR-30e-3p and SERPINE1 expression should be done in multiple HNSCC samples.
- Normal cell line should be used as a control for Sa3 and SAS, at least to show relative expression levels of miR-30e-3p and SERPINE1.
- Line 91-92, in addition to the Kaplan-Meier survival analysis, a cox regression should be performed and report hazard ratio should be mentioned.
- Rationale of choosing miR-30e-3p is not clear from the background.
- Clarify the difference between mock and controls in the figure panels.
Minor comments
- Results section “SERPINE1-mediated pathways in HNSCC cells assessed by gene set enrichment analysis”, the reviewer would caution the authors for making any claims based only on the pathway enrichment analysis. For example, unless EMT markers are tested experimentally, “EMT” as an enriched pathway probably does not deserve a separate results sub-section.
- miR-30 family members have been studied quite extensively in HNSCC, please provide relevant references in the introduction.
- The authors mentioned in the discussion, “Recently, an interesting study of miR-30e-3p reported that miR-30e-3p possessed two functions (tumor-suppressor or oncogene) depending on TP53 status". With wild-type TP53, miR-30e-3p targeted MDM2, and it seems to behave as a tumor suppressor. In contrast, with a nonfunctional TP53, miR-30e-3p behaved as an oncogene. Is this cancer specific, or does it hold true in HNSCC samples, did you check the p53 status in HNSCC samples?
Author Response
International Journal of Molecular Sciences
Revise letter (ijms-1582674)
International Journal of Molecular Sciences
Special Issue: “Advances in Molecular Mechanism of Head and Neck Cancer”
March 13, 2022
Dr. Nijiro Nohata
Guest Editor
Dear Dr. Nohata,
We would like to express our gratitude for your consideration of our above-mentioned manuscript for publication in International Journal of Molecular Sciences. Enclosed, please find the revised manuscript (ijms-1582674) along with a detailed explanation of the revisions, which were made based on the reviewers’ comments. All changes are highlighted in the revised manuscript.
Reviewer #1
Major comments: The main concern of this reviewer is novelty. There are reports of miRNAs (for example, miR-617, miR-181c-5p) targeting SERPINE1 in two of the most common forms of HNSCC, oral squamous cell carcinoma (Zhao and Liu, mol cell biol, 2021) and Laryngeal squamous cell carcinoma (Li X, Front. Oncol., 2020), respectively. Additionally, SERPINE1 is a well-established oncogene in several cancers including HNSCC (Pavon MA, oncotarget, 2015).
One major experiment Minemura C et al., have done is the downregulation of SERPINE1 expression in HNSCC cell which the first two papers have not done, still according to this reviewer, the manuscript in its current format is not a substantial advancement in the field of HNSCC.
Response: I would like to thank you for your important comment. I referred and discussed the articles you pointed out as follows.
(Discussion last chapter)
Accumulating studies have shown that miRNAs regulated the malignancy of cancer cells by controlling the target genes [54]. Downregulation of tumor-suppressive miRNAs cause to aberrant expression of oncogenic genes in cancer cells. The involvement of several tumor-suppressive microRNAs has been reported as a caused of overexpression of SERPINE1 in cancer cells [52]. Previous study showed that miR-617 was directly bound in the UTR of SERPINE1 mRNA, and controlled its expression in OSCC cells [55]. Other study demonstrated that ectopic expression of miR-181c-5p suppressed expression of SERPINE1 in HNSCC cells [56]. Notably, expression of miR-617 and miR-181c-5p inhibited cancer cell proliferation, migration abilities in OSCC/HNSCC cells [55, 56].
Apart from the above, there are several other comments/concerns that needs to be addressed.
Major Comments
Comment-1: Validation of the TCGA analysis for miR-30e-3p and SERPINE1 expression should be done in multiple HNSCC samples.
Response: Following the reviewer’s comment, we attempted to analyzed expression levels of miR-30e-3p and SERPINE1 using other dataset. The results are shown in Supplemental Figure 1,3,4 and following explanatory text has been added.
(Results 2.1 chapter)
Downregulation of miR-30e-3p in HNSCC tissues was confirmed by other datasets (GSE45238 and GSE31277; Supplemental Figure 1).
(Results 2.4 chapter)
To confirm upregulation of these targets genes in HNSCC tissues, we verified using other datasets (GSE30784 and GSE59102; Supplemental Figure 3, 4).
(Material and Methods 4.6 chapter)
To validate the expression of miR-30e-3p and identified target genes in multiple HNSCC samples, GSE45238, GSE31277, GSE30784, and GSE59102 were downloaded from GEO datasets.
Comment-2: Normal cell line should be used as a control for Sa3 and SAS, at least to show relative expression levels of miR-30e-3p and SERPINE1.
Response: We are fully aware of the importance of reviewer comment. However, we do not have a suitable cell line derived from normal tissues derived from the oral cavity and head and neck. We are very sorry, but we are unable to present the data you requested.
Comment-3: Line 91-92, in addition to the Kaplan-Meier survival analysis, a cox regression should be performed and report hazard ratio should be mentioned.
Response: As suggested by the reviewer’s comment, Monovaritate analyses were performed and Kaplan-Meier survival analyses in Figures 1 and 4 were modified. Following text has been added to Results 2.1 and Figure legend of Figure 1.
[log lank p value=0.0353, hazard ratio (HR)=0.6097, 95% confidence interval (95% CI): 0.3828-0.9711]
Comment-4: Rationale of choosing miR-30e-3p is not clear from the background.
Response: We agree with the reviewer’s comment. The following text has been added to the Abstract and Introduction.
(Abstract)
Recent our studies revealed that some passenger strands of microRNAs (miRNAs) closely involved in cancer pathogenesis. Analysis of miRNA expression signatures showed that expression of miR-30e-3p (the passenger strand of pre-miR-30e) was significantly downregulated in caner tissues.
(Introduction 5th chapter)
We have created the miRNA expression signatures in various types of cancers [17,22,23]. Analysis of our miRNA signatures and other studies revealed that some member of miR-30 family were frequently downregulated in cancer tissues, suggesting miR-30 family acted as pivotal tumor-suppressors [24-27]. The Cancer Genome Atlas (TCGA) datasets analysis showed that miR-30e-3p (the passenger strand derived from pre-miR-30e) was significantly downregulated in HNSCC tissues, and its low expression predicted worse prognosis of the disease.
Comment-5: Clarify the difference between mock and controls in the figure panels.
Response: Following the reviewer’s suggestion, the following text has been added to the Material and Methods 4.1. The added text is highlighted in yellow.
“mock”: transfection reagent only
“control”: negative control miRNA precursor that have no function transfected
Minor comments
Comment-1: Results section “SERPINE1-mediated pathways in HNSCC cells assessed by gene set enrichment analysis”, the reviewer would caution the authors for making any claims based only on the pathway enrichment analysis. For example, unless EMT markers are tested experimentally, “EMT” as an enriched pathway probably does not deserve a separate results sub-section.
Response: I agree with the reviewer’s suggestion. The last chapter of Results (Results 2.7.), I will omit it from the text. Include this part in the Results 2.6 as follows (Modified text were highlighted in yellow). Figure 8 were deleted and only the figure of “EMT” on GSEA analysis was added to Figure 7 as “Figure7D”. The following sentence was added to the figure legend for Figure 7D and highlighted in yellow: (D) GSEA analysis showed that most enrichment pathway was “Epithelial Mesenchymal Transition”
Comment-2: miR-30 family members have been studied quite extensively in HNSCC, please provide relevant references in the introduction.
Response: Following the reviewer's comment, I added some articles, and added the following text in Introduction.
(Introduction 5th chapter)
Analysis of our miRNA signatures and other studies revealed that some members of miR-30 family were frequently downregulated in cancer tissues, suggesting miR-30 family acted as pivotal tumor-suppressors [24-27].
Comment-3: The authors mentioned in the discussion, “Recently, an interesting study of miR-30e-3p reported that miR-30e-3p possessed two functions (tumor-suppressor or oncogene) depending on TP53 status". With wild-type TP53, miR-30e-3p targeted MDM2, and it seems to behave as a tumor suppressor. In contrast, with a nonfunctional TP53, miR-30e-3p behaved as an oncogene. Is this cancer specific, or does it hold true in HNSCC samples, did you check the p53 status in HNSCC samples?
Response: As suggested by the reviewer’s comment, I added following text in Discussion.
It is an interesting finding that the role of miR-30e-3p varies depending on the status of the TP53. It is necessary to examine whether this situation is a universal phenomenon in HNSCC cells.
Thank you for your constructive comments and suggestions. We believe that our manuscript has been greatly improved and is now suitable for publication in IJMS. Again, thank you for your consideration of our manuscript for publication in your journal.
Sincerely yours,
Naohiko Seki, Ph.D.
Reviewer 2 Report
This study examined the function of miR-30e-3p in HNSCC and underlying mechanisms, which provides a potential prognosis marker for HNSCC.
Major:
- Add brief introduction of miR-30e-3p in the abstract.
- How about the expression of miR-30e-5p in TCGA-HNSC specimens? miR-30e-5p would be a good control for the analysis in Figure1.
- Please perform the qPCR and RIP assay wi/wo miR-30e-3p to validate the targets.
- Figure 5C, please do co-staining of SERPINE1 and miR-30e-3p in clinical specimens.
- Figure 7, how about the effect of miR-30e-3p on the function of SERPINE-1? Adding siSERPINE-1+miR-30e-3p deletion/knockdown could address this question.
Minor:
- Please remove the repeated sentence in the abstract. “Our miRNA-based approach will accelerate our understanding of the molecular pathogenesis of HNSCC.”
Author Response
Revise letter (ijms-1582674)
International Journal of Molecular Sciences
Special Issue: “Advances in Molecular Mechanism of Head and Neck Cancer”
March 13, 2022
Dr. Nijiro Nohata
Guest Editor
Dear Dr. Nohata,
We would like to express our gratitude for your consideration of our above-mentioned manuscript for publication in International Journal of Molecular Sciences. Enclosed, please find the revised manuscript (ijms-1582674) along with a detailed explanation of the revisions, which were made based on the reviewers’ comments. All changes are highlighted in the revised manuscript.
Reviewer #2
This study examined the function of miR-30e-3p in HNSCC and underlying mechanisms, which provides a potential prognosis marker for HNSCC.
Major Comments
Comment-1: Add brief introduction of miR-30e-3p in the abstract.
Response: According to the reviewer’s suggestion, we modified abstract as follows.
(Abstract)
Recent our studies revealed that some passenger strands of microRNAs (miRNAs) closely involved in cancer pathogenesis. Analysis of miRNA expression signatures showed that expression of miR-30e-3p (the passenger strand of pre-miR-30e) was significantly downregulated in cancer tissues.
Comment-2: How about the expression of miR-30e-5p in TCGA-HNSC specimens? miR-30e-5p would be a good control for the analysis in Figure 1.
Response: I have been submitting a paper on the miR-30 family of HNSCC cells to other journal. In this paper, we refer to miR-30e-5p as follows.
(Results 2.1 chapter)
The expression of miR-30e-5p (the guide strand derived from pre-miR-30e) was also downregulated in HNSCC tissues by TCGA database analysis (data not shown).
Comment-3: Please perform the qPCR and RIP assay wi/wo miR-30e-3p to validate the targets.
Response: The reagents required for the analysis of RIP assay are not supplied, and the prospect of the experiment is unclear.
So instead, I did the following experiment.
I conducted an experiment on the miR-30e-3p target genes (11 types shown in the Figure 3 and 4) to see if miR-30e-3p suppresses expression levels of these genes in HNSCC cells.
The experimental results are shown in the Supplemental Figure 5 and mentioned as follows. Also, we confirmed the expression levels of these target genes in HNSCC tissues using other datasets (Supplemental Figure 3, 4).
(Results 2.4 chapter)
To confirm upregulation of these target genes in HNSCC tissues, we verified using other datasets (GSE30784 and GSE59102; Supplemental Figure 3, 4).
Furthermore, it was examined by quantitative PCR whether these target genes were regulated by miR-30e-3p in HNSCC cells. It was revealed that the expression levels of all genes were reduced by the miR-30e-3p transfected in HNSCC cells (Supplemental Figure 5). PCR primer sequences were shown in Supplemental Table 3.
(Material and Methods 4.3 chapter)
The TaqMan assays used in this study were summarized in Supplemental Table 1. The primers for SYBR Green assays designed in this study were summarized in Supplemental Table 3.
Comment-4: Figure 5C, please do co-staining of SERPINE1 and miR-30e-3p in clinical specimens.
Response: It is well known that RNA in situ hybridization analysis using clinical samples is a fairly difficult technique.
We cannot show the data you request. We ask your kind understanding and cooperation.
Comment-5: Figure 7, how about the effect of miR-30e-3p on the function of SERPINE-1? Adding siSERPINE-1+miR-30e-3p deletion/knockdown could address this question.
Response: The reagents required for your comment are not supplied (microRNA inhibitor and Matrigel chamber assessed invasion assay). I am very sorry, but the prospect of the experiment is unclear. We ask for your kind understanding and cooperation.
Minor comments
Comment-1: Please remove the repeated sentence in the abstract. “Our miRNA-based approach will accelerate our understanding of the molecular pathogenesis of HNSCC.”
Response: As suggested by the reviewer’s comment, we correct it.
Thank you for your constructive comments and suggestions. Some experiments could not be performed on the reviewer's comments. However, I have made some corrections as pointed out by the reviewers, and the quality is comparable to the papers that have been accepted in your journal “IJMS” so far. Again, thank you for your consideration of our manuscript for publication in your journal.
Sincerely yours,
Naohiko Seki, Ph.D.
Round 2
Reviewer 2 Report
The authors have provided a nicely detailed and thorough response to the comments from the previous review and have addressed my major concerns.